# Evaluation of Motor Coordination and Antidepressant Activities of *Cinnamomum osmophloeum* ct. Linalool Leaf Oil in Rodent Model

**DOI:** 10.3390/molecules26103037

**Published:** 2021-05-19

**Authors:** Hui-Ting Chang, Mei-Ling Chang, Yen-Ting Chen, Shang-Tzen Chang, Fu-Lan Hsu, Chia-Chen Wu, Cheng-Kuen Ho

**Affiliations:** 1School of Forestry and Resource Conservation, National Taiwan University, Taipei 106, Taiwan; r05625034@ntu.edu.tw (Y.-T.C.); peter@ntu.edu.tw (S.-T.C.); 2Department of Food Science, Nutrition, and Nutraceutical Biotechnology, Shih Chien University, Taipei 104, Taiwan; mlchang@g2.usc.edu.tw; 3Taiwan Forestry Research Institute, Council of Agriculture, Executive Yuan, Taipei 100, Taiwan; fulan40@gmail.com (F.-L.H.); chiachen@tfri.gov.tw (C.-C.W.); ckho@tfri.gov.tw (C.-K.H.)

**Keywords:** antidepressant effect, *Cinnamomum osmophloeum*, linalool, motor coordination activity

## Abstract

*Cinnamomum* plants (Lauraceae) are a woody species native to South and Southeast Asia forests, and are widely used as food flavors and traditional medicines. This study aims to evaluate the chemical constituents of *Cinnamomum osmophloeum* ct. linalool leaf oil, and its antidepressant and motor coordination activities and the other behavioral evaluations in a rodent animal model. The major component of leaf oil is linalool, confirmed by GC-MS analysis. Leaf oil would not induce the extra body weight gain compared to the control mice at the examined doses after 6 weeks of oral administration. The present results provide the first evidence for motor coordination and antidepressant effects present in leaf oil. According to hypnotic, locomotor behavioral, and motor coordination evaluations, leaf oil would not cause side effects, including weight gain, drowsiness and a diminishment in the motor functions, at the examined doses. In summary, these results revealed *C. osmophloeum* ct. linalool leaf essential oil is of high potential as a therapeutic supplement for minor/medium depressive syndromes.

## 1. Introduction

Cinnamon plants (*Cinnamomum* species, Lauraceae) are one of the important natural product resources from forests and are used as the well-known spice and ethnopharmacological remedy. *Cinnamomum osmophloeum* Kanehira is one of the traditional medicinal plants; its bark is used as a folk medicine for treating colds and improving gastrointestinal function. Essential oils are distilled from its leaves, bark and seeds for various purposes. The leaves are common ingredients in herbal teas and a classic spice in the rural cuisine. Recent studies have confirmed that natural products from *C. osmophloeum* exhibit antibacterial [1], xanthine oxidase inhibitory [2], insecticidal [3], antioxidant [4], antianxiety [5] activities, etc.

Depression is a mental disorder that involves physical and mental conditions, often with pessimistic moods and negative emotions, causing sleep difficulties and a decline in bodily functions, and it is one of the most important diseases that cannot be ignored. Many studies revealed the usefulness of St John’s wort extract (*Hypericum perforatum*), a commercial antidepressant supplement, which can be used to soothe mild and moderate depression [6,7,8,9,10]. Other specific plants have also shown antidepressant activity and bioactive ingredients, such as *Lavandula officinalis*, *Piper nigrum*, *Bacopa monniera*, *Justicia spicigera* and saffron (*Crocus sativus*) [11,12,13,14,15]. Phytochemical compounds, including hyperforin, curcumin, resveratrol, and cinnamic acid, exhibit antidepressant effects [7,16,17,18].

The psychological and physical symptoms of depressive disorders are melancholy mood, fear, no motivation, changes in appetite, sleep problems, psychomotor retardation, etc. Additionally, common psychological and physical symptoms of anxiety disorders include restlessness, nervousness, irritability, uncontrollable worry, difficulty concentrating, panic, sleep disorders, fatigue, fear, etc. Depression and anxiety may occur simultaneously and have the similar symptoms. There are different signs and symptoms between depression and anxiety, and pharmacological and psychological therapies for anxiety disorders are different from depressive disorders [16,19,20,21].

Previous scientific findings reported that linalool, a common monoterpene present in many aromatic plant essential oils, possesses antianxiety activity, a positive effect on social interaction, and a reduction in aggressive behaviors [5,22,23,24]. Rare research has mentioned the antidepressant activity of linalool, and no paper has investigated the motor coordination activity and the other behavioral effects of linalool. Therefore, the aims of this study were to analyze the chemical constituents of *C. osmophloeum* ct. linalool leaf oil, and to validate its antianxiety activity, locomotor activity, antidepressant effect, antinociceptive activity, hypnotic effect, and motor coordination activity in mice by rodent behavioral assays. It is expected to confirm the antianxiety activity of *C. osmophloeum* ct. linalool leaf oil, and firstly evaluate the antidepressant effect, motor coordination activity and other bioactive properties.

## 2. Results and Discussion

### 2.1. Constituent Analysis of Cinnamomum osmophloeum ct. Linalool Leaf Oil

A schematic illustration of constituent analysis and behavioral evaluations of *Cinnamomum osmophloeum* ct. linalool leaf oil is shown in Figure 1. According to gas chromatography-mass spectrometry (GC-MS) analyses (Figure 2), the main component of the selected *C. osmophloeum* leaf oil was linalool (93.2%), followed by coumarin (2.3%), camphor (1.5%), β-caryophyllene (1.2%), *trans*-cinnamaldehyde (1.0%), and caryophyllene oxide (0.8%) (Table 1). The content of the major compound, linalool, was more than 90%; it revealed that the quality and stability of leaf oils from *C. osmophloeum* ct. linalool are consistent.

### 2.2. Effect of Leaf Oil on the Body Weight of Mice

The body weight of the ICR mice was measured every week during the experimental period. The initial and final body weights of all mice, including the control, positive control (trazodone hydrochloride), and treatment with three doses (100, 200, and 400 mg/Kg) of leaf oil, were in the normal ranges, as shown in Table 2. The body weight gains of the control and positive control mice increased 7.6 ± 2.1 g and 7.1 ± 1.1 g after 6 weeks. The body weight gains of the mice treated with three doses of leaf oil (LO100, LO200, and LO400) were 8.2 ± 1.9 g, 6.8 ± 2.1 g, and 7.6 ± 2.1 g, respectively, with no significant difference between all groups in the statistical analysis. It reveals that *C. osmophloeum* ct. linalool leaf oil intake did not reduce or increase the appetite of mice; leaf oil would not affect the body weight of mice.

### 2.3. Effect of Leaf Oil on the Antianxiety and Locomotor Behavior of Mice

Table 3 shows the effect of leaf oil on antianxiety activity in mice by the open field assay. The number of entries in the central zone of the control mice was 18.00 ± 4.47, which was much lower than that of the mice treated with trazodone hydrochloride at a dose of 100 mg/Kg (31.14 ± 3.76). It revealed that the treatment of trazodone hydrochloride resulted in the mice more frequently entering the central zone instead of staying in the peripheral zone in the open field test box. No significant difference in the number of entries into the central zone were found between control mice and the mice treated with leaf oil at a low dose (100 mg/kg). The number of entries into the central zone of the mice treated with the dose (400 mg/kg) of leaf oil was 31.00 ± 3.87, indicating that the antianxiety activity of leaf oil treatment was comparable with that of the positive control, trazodone hydrochloride, (*p* < 0.05). The number of entries into the central zone of the mice treated with leaf oil was increased and related to the treated doses. The result indicated the leaf oil has an anxiolytic effect in mice, and it is consistent with the previous studies performed by Linck et al. (2010) [22] and Cheng et al. (2015) [5].

The total distance traveled in the open filed test was also calculated to identify the locomotor activity level of the test mice after different treatments, as shown in Table 3. The control mice and the mice treated with leaf oil at a low dose (100 mg/kg) showed a similar performance: the total distance traveled of both groups were 21.28 ± 3.87 and 21.58 ± 2.88 m, respectively. The significant improvement of the total distance traveled was observed in the other three groups (*p* < 0.05); the total distance traveled of these groups ranged from 24.39 to 25.70 m. The mice treated with trazodone hydrochloride (100 mg/kg) travelled the longest distance, which demonstrated the superior locomotor activity. The higher dose of leaf oil (200 and 400 mg/kg) produced a significant increase in the total distance traveled compared to that of the control mice, it reflects that the locomotor activity level of mice was enhanced after treatment with the high-dose leaf oil.

### 2.4. Effect of Leaf Oil on the Antidepressant Behavior of Mice

The forced swimming test is the typical and representative assay used to evaluate the antidepressant activity in rodent animals. Mahmoudi et al. (2015) reported that *Feijoa sellowiana* leaf extract showed the antidepressant activity in the forced swimming assay; leaf extract treatment remarkably decreased the immobility time of the male Swiss albino mice [20]. Shorter immobility time means the treated mice exhibit more vigorous and active behaviors instead of depressive symptoms in the stress environment. The immobility behavior of mice in the forced swimming test was reduced by the administration of *Turnera diffusa* aqueous extract; results confirmed the antidepressant effect of *T. diffusa* aqueous extract [25].

Figure 3 shows the immobility time of the examined mice in the forced swimming test: the latency of control group mice was 109.50 ± 7.13 s. The immobility time of the mice treated with trazodone hydrochloride, a prescription antidepressant, was significantly shorter (39.75 ± 3.69 s) than that of control mice at the dose of 100 mg/kg. The leaf oil also induced the considerable reduction in immobility time compared with the vehicle control, with a statistical difference (*p* < 0.05); the immobility time of the treated mice was 64.50 ± 4.81 s at the dose of 200 mg/kg. The antidepressant behavior of mice can be markedly improved by the oral administration of the *C. osmophloeum* ct. linalool leaf oil.

### 2.5. Effect of Leaf Oil on the Antinociceptive Behavior of Mice

Tail flicking in rodents is a response of pain caused by the heat stimulus to measure the efficacy of antinociceptive activity. The results of the tail flick test are shown in Table 4; the tail flick time of control mice was 3.75 ± 0.38 s. The mice treated with trazodone hydrochloride exhibited a slight decrease in the tail flick time (3.49 ± 0.32 s) without statistical significance (*p* < 0.05). No differences were found between the mice treated with different doses (100–400 mg/Kg) of leaf oil; the tail flick time was between 3.81–4.01 s. The results indicated that both the leaf oil and trazodone hydrochloride did not exhibit the antinociceptive activity for the experimental mice.

### 2.6. Effect of Leaf Oil on the Hypnotic Effect of Mice

Rahimi et al. (2018) reported the hypnotic effect of the fruit and seed extracts of *Lagenaria vulgaris* on pentobarbital-induced sleep in mice [26]. The extracts of *L. vulgaris* significantly increased the sleeping duration of mice at a dose of 200 mg/kg, the result revealed that the fruit and seed extracts of *L. vulgaris* exhibited the sleep-prolonging effect.

The results of the primidone-induced sleeping test is listed in Table 4; the duration (sleeping time) of the control mice was 55.08 ± 4.72 min. The mice treated with leaf oil at 100 mg/kg showed a slight reduction in the sleeping time (49.91 ± 4.59 min) compared to the control, but the data of both groups were not statistically significant. The durations of the other treated mice were all lower than that of the control group, indicating that no hypnotic activity was observed for leaf oil in the primidone-induced sleeping test in none of the examined doses. Some antidepressant medicines would cause drowsiness to different extents in patients. Results from the hypnotic evaluation revealed that *C. osmophloeum* ct. linalool leaf oil would not lead to drowsiness at the examined doses.

### 2.7. Effect of Leaf Oil on the Motor Coordination Behavior of Mice

Motor coordination activity was evaluated by the rotarod assay, as depicted in Figure 4. The duration time of the control mice remained on the rotating rod was 192.40 ± 19.03 s. There was no significant difference in latency to falling in rotarod between the control mice and the mice treated with the low dose of leaf oil (100 mg/kg). The mice treated with the higher doses of leaf oil (200 and 400 mg/kg) showed a large increase in the time spent on the rotating rod, as compared with the positive control, trazodone hydrochloride, at the dose of 100 mg/kg (*p* < 0.05); the duration time of these group were 293.71 ± 12.19, 290.67 ± 14.68, and 286.50 ± 22.05 s, respectively. A maximum improvement in the duration time of the mice treated leaf oil (200 mg/kg) was 52.66% compared to the control group.

Muthaiyah et al. (2014) reported that dietary walnuts can effectively improve the psychomotor coordination in a transgenic mouse model of Alzheimer’s disease; it suggested dietary supplementation of natural products is of benefit to enhance the motor coordination skill [27]. Rekha et al. (2013) investigated the motor coordination effect of geraniol, an acyclic monoterpene alcohol; the results displayed that geraniol exhibited the neuroprotective effect to alleviate the chemically induced neurotoxicity and improve the motor coordination of treated mice in the rotarod assay [28]. Linalool is a structural isomer of geraniol; both compounds possess the double bond and alcoholic functional groups. The precise mechanism and neuroprotective effect of linalool need further investigation.

The above result reveals that the balance and motor activity of mice can be significantly enhanced through the inhalation of *C. osmophloeum* ct. linalool leaf oil. It is in accordance with previous findings from the locomotor behavioral and hypnotic effect evaluations. This is the first study to report the motor coordination activity of *C. osmophloeum* ct. linalool leaf oil.

## 3. Materials and Methods

### 3.1. Hydrodistillation of Leaf Oil

Fresh leaves (22.78 Kg) of *Cinnamomum osmophoeum* Kanehira ct. linalool were harvested from the Taimalee Research Center of Taiwan Forestry Research Institute, Taitung, Taiwan. The identification of *C. osmophoeum* was authenticated in the Silviculture Division, Taiwan Forestry Research Institute by Dr. Cheng-Kuen Ho. Leaves were hydrodistilled in the conventional apparatus in batches to obtain total 176 mL of leaf essential oil, and the yield of leaf oil was ca. 3%. Leaf oil was kept in dark glass bottles and stored in a refrigerator at 4 °C.

### 3.2. GC-MS Analysis of Leaf Oil

The chemical constituents in leaf oils was analyzed by using the gas chromatography-mass spectrometer (GC-MS, GCMS-QP2010 Ultra, Shimadzu, Kyoto, Japan) equipped with a DB-5MS capillary column (Crossbond 5% phenyl methylpolysiloxane, 30 m length × 0.25 mm i.d. × 0.25 µm film thickness, J&W Scientific, Folsom, CA, USA). The temperature of the injection port was set at 250 °C. The oven temperature was initially held at 60 °C for 3 min, then increased to 150 °C at a rate of 3 °C/min, and finally increased to 240 °C at a rate of 5 °C/min and held for 5 min. The carrier gas was helium at a flow rate of 1 mL/min, the split ratio was 1:10, and the GC/MS interface temperatures were set at 250 °C. Constituents were identified by comparing mass spectra (m/z 50–650 amu) with the National Institute of Standards and Technology (NIST) and Wiley library databases, the arithmetic index (AI) [29], and authentic standards of constituents. The quantification of constituents was obtained by integrating the peak area of the chromatogram by GC coupled to a flame ionization detector (FID).

### 3.3. Animals and Treatments

Adults male ICR mice (weighing 30–38 g) were housed under constant room temperature (22–25 °C) and inverted light–dark cycle (12:12 h) with free access to water and food ad libitum. The mice were allowed to adapt to the laboratory environment for 1 week and were randomly divided into five groups (*n* = 7). The leaf oil (LO) was evenly mixed in soybean oil, and was administered daily by oral route (p.o.) via gavage to the animals at doses of 100, 200 and 400 mg/kg, respectively, using soybean oil as the vehicle (control, soybean oil). The positive control was trazodone hydrochloride (TH, 100 mg/kg), a prescription antidepressant approved by the FDA. Leaf oil and trazodone hydrochloride were administered each day over the whole experimental period. The body weight of each mouse was recorded once a week. After 2 weeks of treatment with leaf oil, behavioral tests were performed between 8 a.m. and 5 p.m. All behavioral protocols were approved by the Institutional Animal Care and Use Committee (IACUC) in National Taiwan University (IACUC Approval No: NTU105-EL-00088), and all mice were handled by the 3R principles of laboratory animal care and use.

### 3.4. Open Field Test (Antianxiety and Locomotor Behavioral Evaluation)

The open field test is a classic assay used to evaluate the anxiety and locomotor activities in mice. One hour after oral administration, the mice were placed in the central zone of an open box (60 (L) × 60 (W) × 40 (H) cm). The bottom area of the test box is divided into 16 square blocks, the middle 4 blocks is the central zone, the peripheral zone is 12 outer blocks. When the mouse enters the new environment, the mouse will spontaneously approach the outer perimeter near the wall area. A digital imaging system was used to analyze the number of entries in the central zone (antianxiety effect) and total travel distance (locomotor activity) during the 5 min of the test period [30,31,32].

### 3.5. Forced Swimming Test (Antidepressant Behavioral Evaluation)

According to the related studies [20,33,34,35], the essential oil was administered one hour before the forced swimming test (FST), the test mouse was individually placed in an open cylinder (25 cm in high and 10 cm in diameter) with a water depth of 15 cm held at 25 °C. A video camera was used to record each experimental process for 6 min, and immobility time was defined as the time the mouse spent floating in the open cylinder without struggling and slightly treading to keep the head above water during the final 4 min of the test. The decrease in immobility time indicates the antidepressant-like behavior of treated mice.

### 3.6. Tail Flick Test (Antinociceptive Behavioral Evaluation)

After oral administration for 1 h, the test mice were placed in the rodent restrainer, a plastic tube with ventilation. One-third of the tail of the test mouse were immersed in water at 50 °C, and then the span of time in seconds was recorded as the tail of mouse was withdrawn from the hot water. When the span of time is more than 10 s, it is recommended to quickly remove the mouse’s tail from hot water to avoid injury to the mice. The longer span of time in the tail flick test reveals the better antinociceptive effect of the treatment [36,37,38].

### 3.7. Primidone-Induced Sleeping Test (Hypnotic Effect Evaluation)

Referring to the related studies [35,39], the primidone-induced sleeping test was conducted to evaluate the hypnotic effect of *C. osmophloeum* ct. linalool leaf oil in mice. After the oral administration of each treatment for 1 h, primidone (an analog of phenobarbital) was injected into mice at a dose of 30 mg/kg (intraperitoneal injection, i.p.) to induce sleep. The duration of sleep of each mouse was recorded; the longer duration indicates the better hypnotic effect of the treatment.

### 3.8. Rotarod Test (Motor Coordination Evaluation)

Motor coordination activity was evaluated by the rotarod apparatus (LE8205, Panlab, 5 slots) consisting of rotating rods (3 cm in diameter). Referring to the related reports [40,41,42], all mice were pre-trained once for 5 min one day before the formal testing. After administering the specimens for 1 h, the mice were placed in the rotarod, and the apparatus would automatically record the latency (duration of time spent on the rotating rod) when each mouse fell from the rod rotating at 10 rpm in a 5-min test session. The longer latency time indicates the better motor coordination ability of the examined mice.

### 3.9. Statistical Analysis

All data were presented as mean values and standard deviations. The significance of difference was analyzed using the Scheffe’s test, a completely post hoc multiple comparison analysis with stringent error control, of the SAS software (version 9.2, Cary, NC, USA) with a 95% confidence interval.

## 4. Conclusions

The study demonstrated the behavioral effects of *C. osmophloeum* ct. linalool leaf oil after oral administration in mice, the motor coordination and antidepressant activities of leaf oil were firstly reported by a rodent model system, to the best of our knowledge. In the open field test, the oral treatment on mice with leaf oil also increased the number of entries in the central zone and total travel distance; it provided evidence that the leaf oil exhibited anxiolytic and locomotor activities. The oral treatment on mice with leaf oil decreased the immobility time and induced the antidepressant effect in the forced swimming test. The motor coordination activity of mice was improved after the administration of leaf oil, and the time spent on the rotating rod of the treated mice was significantly increased in the rotarod assay. These results revealed that the *C. osmophloeum* ct. linalool leaf oil has antidepressant and anxiolytic activities, enhances the locomotor and motor coordination effects, and has the promising potential to be a therapeutic supplement of minor/medium depressive syndromes.

## Figures and Tables

**Figure 1 molecules-26-03037-f001:**
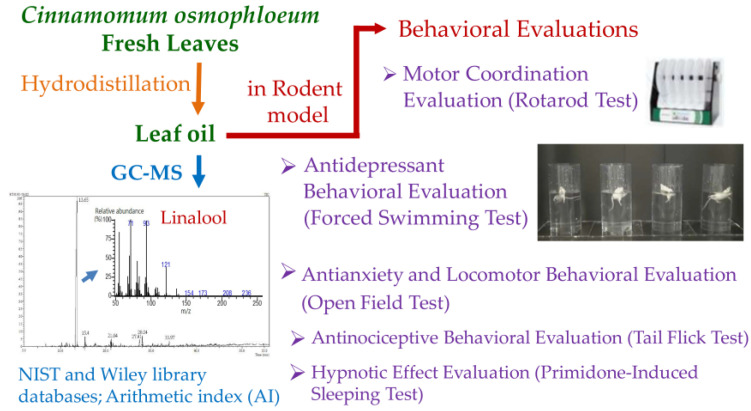
Schematic illustration of constituent analysis and behavioral evaluations of *C. osmophloeum* ct. linalool leaf oil.

**Figure 2 molecules-26-03037-f002:**
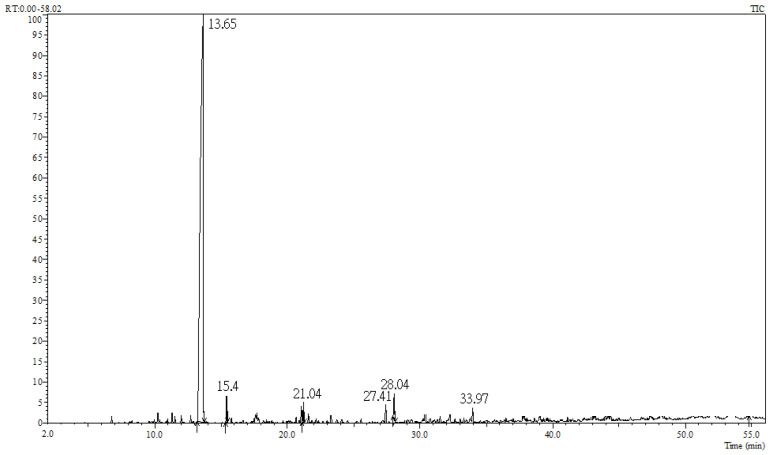
Gas chromatogram of *C. osmophloeum* ct. linalool leaf oil.

**Figure 3 molecules-26-03037-f003:**
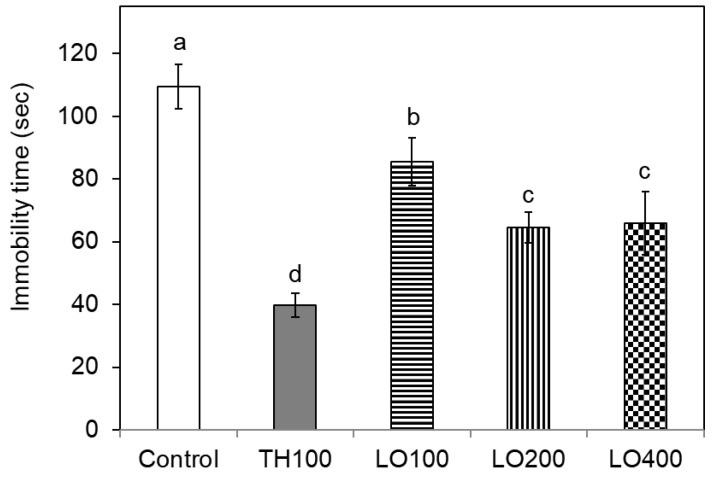
Effect of different doses of *C. osmophloeum* ct. linalool leaf oil on antidepressant activity in ICR mice by the forced swimming assay. Data are presented as mean ± S.D. (*n* = 7); TH100: trazodone hydrochloride, 100 mg/Kg; LO100, LO200, and LO400: leaf oil, 100 mg/Kg, 200 mg/Kg, and 400 mg/Kg. Different letters (a–d) in the figure are significantly different at the level of *p* < 0.05 according to the Scheffe’s test.

**Figure 4 molecules-26-03037-f004:**
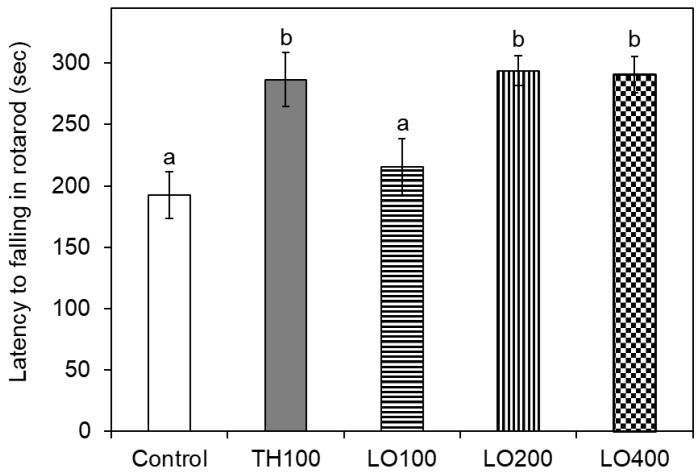
Effect of different doses of *C. osmophloeum* ct. linalool leaf oil on motor coordination in ICR mice by the rotarod assay. Data are presented as mean ± S.D. (*n* = 7); TH100: trazodone hydrochloride, 100 mg/Kg; LO100, LO200, and LO400: leaf oil, 100 mg/Kg, 200 mg/Kg, and 400 mg/Kg. Different letters (a–b) in the figure are significantly different at the level of *p* < 0.05 according to the Scheffe’s test.

**Table 1 molecules-26-03037-t001:** Constituents of *C. osmophloeum* ct. linalool leaf oil.

RT	AI	Compound	Formula	Relative Content (%)	Identification Method
13.65	1091	Linalool	C_10_H_18_O	93.2	MS, AI
15.40	1141	Camphor	C_10_H_16_O	1.5	MS, AI
21.04	1264	*trans*-Cinnamaldehyde	C_9_H_8_O	1.0	MS, AI
27.41	1418	β-Caryophyllene	C_15_H_24_	1.2	MS, AI
28.04	1431	Coumarin	C_9_H_6_O_2_	2.3	MS, AI
33.97	1580	Caryophyllene oxide	C_15_H_24_O	0.8	MS, AI

RT: retention time (min); AI: arithmetic index relative to *n*-alkanes (C8-C30) on a DB-5MS column.

**Table 2 molecules-26-03037-t002:** Effect of different doses of *C. osmophloeum* ct. linalool leaf oil on the body weight gain of ICR mice after 6 weeks of oral administration.

	Control	TH100	LO100	LO200	LO400
Initial body weight (g) *	30.1 ± 1.6	30.4 ± 2.1	30.3 ± 1.5	30.7 ± 1.5	30.1 ± 2.1
Body weight after 6 weeks (g) *	37.9 ± 2.3	37.2 ± 2.2	37.9 ± 2.1	38.1 ± 3.2	37.1 ± 3.1
Weight gain (g/6 weeks) *	7.6 ± 2.1	7.1 ± 1.1	8.2 ± 1.9	6.8 ± 2.1	7.6 ± 2.1

Data are presented as mean ± S.D. (*n* = 7). * Values in the table are not significantly different at the level of *p* < 0.05 according to the Scheffe’s test. TH100: trazodone hydrochloride, 100 mg/Kg; LO100, LO200, and LO400: leaf oil, 100 mg/Kg, 200 mg/Kg, and 400 mg/Kg.

**Table 3 molecules-26-03037-t003:** Effect of different doses of *C. osmophloeum* ct. linalool leaf oil on antianxiety activity in ICR mice by the open field assay.

Specimen	Number of Entries into Central Zone	Total Distance Traveled (m)
Control	18.00 ± 4.47 ^a^	21.28 ± 2.11 ^A^
TH100 *	31.14 ± 3.76 ^c^	25.70 ± 2.56 ^B^
LO100	20.71 ± 4.79 ^a,b^	21.58 ± 2.88 ^A^
LO200	28.71 ± 5.22 ^b,c^	25.30 ± 2.62 ^B^
LO400	31.00 ± 3.87 ^c^	24.39 ± 2.33 ^B^

Data are presented as mean ± S.D. (*n* = 7). *: Positive control; TH100: trazodone hydrochloride, 100 mg/Kg; LO100, LO200, and LO400: leaf oil, 100 mg/Kg, 200 mg/Kg, and 400 mg/Kg. Different letters (a–c and A–B) in the table are significantly different at the level of *p* < 0.05 according to the Scheffe’s test.

**Table 4 molecules-26-03037-t004:** Effect of different doses of *C. osmophloeum* ct. linalool leaf oil on antinociceptive and hypnotic activities in ICR mice.

Specimen	Time (sec) in the Tail Flick Test *	Duration (min) in the Primidone-Induced Sleeping Test *
Control	3.75 ± 0.38	55.08 ± 4.72
TH100	3.49 ± 0.32	49.87 ± 5.36
LO100	3.91 ± 0.12	49.91 ± 4.59
LO200	3.81 ± 0.28	50.10 ± 4.22
LO400	4.01 ± 0.19	52.10 ± 5.61

Data are presented as mean ± S.D. (*n* = 7). TH100: trazodone hydrochloride, 100 mg/Kg; LO100, LO200, and LO400: leaf oil, 100 mg/Kg, 200 mg/Kg, and 400 mg/Kg. * Values in the table are not significantly different at the level of *p* < 0.05 according to the Scheffe’s test.

## Data Availability

The data are available from the corresponding author on reasonable request.

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
