# Peer review of "Evaluation of Motor Coordination and Antidepressant Activities of Cinnamomum osmophloeum ct. Linalool Leaf Oil in Rodent Model"

_molecules, 2021, doi:10.3390/molecules26103037_

Round 1

Reviewer 1 Report

The manuscript presents interesting results, which were obtained by classical pharmacology techniques. It is necessary to present the botanical identification of the species studied.

Author Response

Response to Reviewer 1 Comments

Point 1: The manuscript presents interesting results, which were obtained by classical pharmacology techniques. It is necessary to present the botanical identification of the species studied.

Response 1: The botanical identification of the species studied has been added in the text (Page 7, paragraph 3.1.) of revised manuscript. Thank you very much for your valuable suggestion.

Reviewer 2 Report

The following manuscript with the title "Evaluation of Motor Coordination and Antidepressant Activities of Cinnamomum osmophloeum ct. Linalool Leaf Oil in Rodent Model" is well prepared. Introductory story goes very well. Results and discussion section is presented in a good manner, including presentation of tables and figures. Material and methods are clearly presented and conventional methodology is applied for this type of the study.

I would suggest minor corrections of the manuscript before publication. Please correct typographical mistakes throughout the manuscript; eg. first sentence in introduction Cinnamomum should be italicized in full not like Cinnamomum etc. Some sentences could be better constructed especially in results section.

Author Response

Response to Reviewer 2 Comments

Point 1: The following manuscript with the title "Evaluation of Motor Coordination and Antidepressant Activities of Cinnamomum osmophloeum ct. Linalool Leaf Oil in Rodent Model" is well prepared. Introductory story goes very well. Results and discussion section is presented in a good manner, including presentation of tables and figures. Material and methods are clearly presented and conventional methodology is applied for this type of the study.

I would suggest minor corrections of the manuscript before publication. Please correct typographical mistakes throughout the manuscript; eg. first sentence in introduction Cinnamomum should be italicized in full not like Cinnamomum etc. Some sentences could be better constructed especially in results section.

Response 1: The typographical mistakes and sentences have been corrected and constructed throughout the manuscript. Thank you very much for your valuable suggestions.

Reviewer 3 Report

The submitted manuscript by Chang et al. is focussed on evaluation of several effects the leaf oil from Cinnamonum osmophloeum containing linalool in > 93 %. The minor components of the extract were also identified. The investigation is well planned and conducted. All necesasary data were supplied. The results are clearly presented and discussed, and the conclusions are adequately concise.

Minor formal error found:

Page 7, paragraph 3.5., the first line of the paragraph: the word "administrated" should be replaced by "administered".

Author Response

Response to Reviewer 3 Comments

Point 1: The submitted manuscript by Chang et al. is focussed on evaluation of several effects the leaf oil from Cinnamomum osmophloeum containing linalool in > 93 %. The minor components of the extract were also identified. The investigation is well planned and conducted. All necessary data were supplied. The results are clearly presented and discussed, and the conclusions are adequately concise.

Minor formal error found:

Page 7, paragraph 3.5., the first line of the paragraph: the word "administrated" should be replaced by "administered".

Response 1: Page 7, paragraph 3.5., the word "administrated" has been replaced by "administered". Thank you very much for your valuable suggestion.

Round 2

Reviewer 1 Report

The authors followed the proposed suggestions and the manuscript is in conditions for publication.